# A Case Study in Saudi Arabia: Biodiversity of Maize Seed-Borne Pathogenic Fungi in Relation to Biochemical, Physiological, and Molecular Characteristics

**DOI:** 10.3390/plants11060829

**Published:** 2022-03-21

**Authors:** Abdulaziz A. Al-Askar, Khalid M. Ghoneem, Elsayed E. Hafez, WesamEldin I. A. Saber

**Affiliations:** 1Botany and Microbiology Department, Faculty of Science, King Saud University, Riyadh 11451, Saudi Arabia; 2Seed Pathology Research Department, Plant Pathology Research Institute, Agricultural Research Center, Giza 12619, Egypt; khalid_ghoneem@yahoo.com; 3Plant Protection and Biomolecular Diagnosis Department, Arid Lands Cultivation Research Institute, City of Scientific Research and Technological Applications, New Borg El-Arab City 21934, Egypt; elsayed_hafez@yahoo.com; 4Microbial Activity Unit, Microbiology Department, Soils, Water and Environment Research Institute, Agricultural Research Center, Giza 12619, Egypt

**Keywords:** enzymes, amino acids, fungal diversity, environmental conditions, phylogeny, maize

## Abstract

Microbiodiversity is usually correlated with environmental conditions. This investigation is a case study to cover the lack of knowledge on the correlation of biochemical, physiological, and molecular attributes with the distribution of seed-borne pathogenic fungi of maize under the environmental conditions of the Kingdom of Saudi Arabia to help forecast any destructive epidemics. Forty-one fungal species belonging to 24 genera were detected using standard moist blotter (SMB), deep freezing blotter (DFB), and agar plate (AP) techniques. SMB was superior in detecting the maximum numbers (36 species) of seed-borne mycoflora. The pathogenicity assay revealed that, among 18 seed-borne fungal pathogens used, 12 isolates caused high percentages of rotted seeds and seedling mortality symptoms, which were identified molecularly using an internal transcribed spacer sequence. Two *Curvularia* spp. and *Sarocladium zeae* were reported for the first time in KSA. The strains showed various enzymatic activities and amino acid profiles under different environmental setups. Temperature and humidity were the environmental variables influencing the fungal pathogenicity. The highest pathogenicity was correlated with the presence and concentration of threonine, alanine, glutamic, aspartic acids, and protein. The study concluded with the discovery of four new phytopathogens in KSA and, further, evidenced a marked correlation among the investigated variables. Nevertheless, more studies are encouraged to include additional physiological properties of the phytopathogens, such as toxigenic activity, as well as extend the fungal biodiversity study to other plants.

## 1. Introduction

Maize (*Zea mays*, L.) is the third-most important crop following wheat and rice and therefore considered the primary staple cereal for food and livestock feed, in addition to uses in the industry for oils and biofuel production as alcohol [1,2]. Maize originated from Americans and was then cultivated in different agroecological zones with a wide range of environmental conditions around the world [3].

According to the environmental conditions and climatic factors, diverse pathogenic fungi were reported on maize grains, causing malformation, growth reduction and decreasing of photosynthetic capacity, and deficiency in nutritional elements due to the depletion of nitrogen, carbon, and other inorganic salts from the host [4]. These changes reduce the quality, quantity, and nutritional values of grain. Moreover, pathogenic fungi produce mycotoxins (aflatoxins and ochratoxins) that are, in addition, harmful to humans and animals [5]. Moreover, since grains are a good substrate for microbial growth, many species of toxicogenic fungi, e.g., *Aspergillus, Penicillium,* and *Curvularia*, and the *Fusarium* genera infect maize and were stated to cause serious diseases such as seed, root, stem, ear, and kernel rots [6,7].

It was found that fungi can distribute and survive in different wide types of ecosystems [8]. The biodiversity and populations of the fungal pathogens correlate with biotic factors like the plant and/or microbial community and abiotic elements such as temperature, humidity, moisture, salinity, and pH of the soil. Both biotic and abiotic factors usually affect also the pathogenicity of the phytopathogenic fungi [9,10].

Little of the environmental literature has taken care of seed-borne mycobiomes. For instance, *Fusarium* ear rot, caused mainly by *Fusarium verticillioides* and *F. proliferatum,* is an important seed-borne pathogenic fungi of maize. Both fungi can survive in infected maize grains and have a wide distribution in temperate regions [11,12]. The moisture content and temperature stress during plant field growth and postharvest are the most related factors that affect fungal growth, spreading, and mycotoxins production by both pathogens on maize grains [5,11,13]. In this connection, maize seed-borne *Fusarium graminearum* and *Stenocarpella maydis* pathogens were reported at a higher incidence in the wetter seasons of the studied sites. It was concluded that the inoculum levels of each of the fungal species can be affected by local factors like the microenvironment in field conditions, as well as land preparation, previous crops, the volume of debris from the preseason, and fertilizer application, which have a bigger impact compared to the drought condition on the fungal community structure of ear rot pathogens [14].

However, the lytic activity of enzymes (e.g., cellulases, amylases, pectinases, and proteases) of the phytopathogenic fungi is found to be linked to the pathogenicity features and therefore play a vital role in disease incidence and development. These enzymes could be used as a differentiating tool to study the biodiversity of phytopathogens [15].

The fungal cell wall supports and protects internal structures and organelles. Protein constitutes 20–30% of the cell wall and is firmly intertwined within the glucan and chitin-based matrix of the filamentous fungi [16]. Among the various components, the protein content mediates molecule absorption, aids in adhesion for cell migration and fusion, maintains the cell shape, protects the cell against irrelevant substances, mediates the molecule’s absorption, transmits intracellular signals from exterior stimuli, and synthesizes and remodels the cell wall components [17]. These features represent a protective barrier against various adverse ecosystems and could therefore be used as an indicator for the survivability of the fungal pathogens under certain ecosystem conditions.

During the pathogenesis, to survive and cause the disease, the fungus produces specialized proteins to suppress the plant’s immune response [18]. In general, amino acids are considered as fuel for the defense genes in a plant, and fuel for the fungus during infection as well; a battle may arise between the fungus and the plant, provided that either of them acquires a large number of amino acids. If the plant contains many amino acids, its resistance to infection is higher [19,20]. Moreover, if the fungus possesses a large number of amino acids, its ability to infect is high; contrarily, a lack of proteins in the plant facilitates the infection process, and their deficiency in the fungus reduces the ability of the fungus to cause the disease [21,22]. That is why the current study investigated both amino acids and proteins of the pathogenic fungi.

Molecular characterization is another tool that uses DNA sequencing for the identification of fungal groups and, in comparison, for the description of fungal biodiversity. Genetic markers as genes associated with internal transcribed spacer (ITS) sections and the highly conserved 5.8S gene between ITS1 and ITS2 are used for taxonomic identification and phylogenetic relationships among fungal communities [23,24].

To the authors’ knowledge, the current pioneer study is the first report on the biodiversity of maize grain-borne pathogenic fungi in Saudi Arabia, representing a case study to draw the attention of the biodiversity of phytopathogenic fungal communities associated with *Zea mays* grain in relation to the pathology, enzymatic activity, cell wall protein content, and molecular identifications. This may enable understanding the ecobiological relationships of these pathogens to facilitate the management process and get rid of them in the future.

## 2. Materials and Methods

### 2.1. Meteorological Data of the Studied Area

Except for Asir Province, the Kingdom of Saudi Arabia region is generally dominated by a dry desert climate associated with extreme daytime temperatures, an abrupt drop in temperatures, and very low annual rainfall. The kingdom is generally hit by dry winds, and almost all the areas are characterized by drought. The Asir region located along the western coast is impacted by Indian Ocean monsoons during October and March, usually accompanied by rainfall drops with an average of 300 mm, which accounts for 60% of the annual precipitation. The temperatures and humidity are highly differentiated as a result of the impact of the tropical and subtropical high-pressure systems, where there is a significant difference between the coastal and intra-tropical areas [25]. During the sampling period (October and November of the 2017 season) for the areas studied, the minimum air temperatures ranged from −2 up to 28 °C, and the maximum was from 18 up to 51 °C, with an average temperature was 25 ± 2 °C. The studied areas had varied relative humidity, from 3% to 100%, with a rainfall rate less than 115 mm per year. The 60 surveyed sites of maize cultivation area represent the diverse climatic conditions in Saudi Arabia, including the Riyadh, Aseer, Al-Ahsaa, Najran, Wadi Al-Dawasir, Al-Jouf, Tabuk, Al Kharj, Gazan, Al-Madinah, Al-Qaseem, and Hail governorates. At least five samples were collected from each site. The investigated area was located between the latitudes of 20°241′ N and 30°683′ N and longitudes of 36°294′ E and 50°474′ E. For each sampling site, the location was georeferenced using the global positioning system (GPS). The coordinates were used to display the spatial positions on the map (Figure 1) that were created using ArcGIS software, version 10.1.

### 2.2. Sampling Process

A total of 60 samples of maize were collected, covering 12 various crop-growing provinces in the kingdom. Each sample collection was obtained from a 50 × 50 m area around each sampling location in a random zigzag pattern. The mature maize corncobs collected were placed in cotton bags, numbered, and stored at 4 °C. Maize seeds of each sample were prepared for seed health testing study by extracting from 100 corncobs and spread out to dry on a porcelain plate for a few days at ambient temperature (25 ± 2 °C).

### 2.3. Inspection of Seed-Borne Fungi

The seed health testing was performed based on the approved methods of the International Seed Testing Association [26], including the standard moist blotter (SMB), deep freezing blotter (DFB), and agar plate (AP) methods. The individual samples were surface-sterilized by soaking for 3 min in 1% sodium hypochlorite (NaClO), washing with sterile water, and spreading them out to dry at an ambient temperature (25 ± 2 °C). A total of 400 seeds were used for each technique. The recovery of each fungal species was evaluated by calculating the percentage of occurrence by each technique.

For the SMB and DFB techniques, the maize seeds were placed inside 9-cm-diameter sterile Petri dishes containing 3 layers of wet blotting paper at 10 grains/Petri dish. Concerning the AP technique, the seeds were plated on potato dextrose agar (PDA) medium, pH 6.5. The plates of both the SMB and AP techniques were then incubated at 20 ± 2 °C for 7 days below cool white fluorescent light with a rotating cycle of light and darkness every 12 h. For DFB, the plates were incubated at 20 ± 2 °C for 24 h and then relocated to a −20 °C freezer for 24 h, followed by a 5-day incubation (20 ± 2 °C) under cool white fluorescent light with rotating cycles of light and darkness every 12 h. Seven days after incubation, the recovered fungi in each plate technique were identified by examining their habit characters under a light-supported stereoscopic microscope.

### 2.4. Purification and Morphological Identification 

Hyphal tip and/or single-spore isolation methods were used for the fungal purification, then transferred to plates containing PDA medium supplemented with an antibacterial agent (L-chloramphenicol 0.1 mg/L and streptomycin sulfate 0.3 mg/L). Pure cultures of the developing fungi were transferred to slants containing potato carrot agar medium for more studies. After purification, the frequency and incidence (mean of the sample infections, I %) for the individual fungi were computed based on the following Equations (1) and (2):(1)$$\mathrm{Frequency}\;\mathrm{of}\;a\;\mathrm{fungus}\;\left(\%\right)=\frac{\mathrm{Number}\;\mathrm{of}\;\mathrm{positive}\;\mathrm{samples}\;\mathrm{of}\;\mathrm{the}\;\mathrm{fungus}}{\mathrm{The}\;\mathrm{total}\;\mathrm{number}\;\mathrm{of}\;\mathrm{tested}\;\mathrm{samples}\;}$$
(2)$$\mathrm{Incidence}\;\mathrm{of}\;a\;\mathrm{fungus}\;\left(\%\right)=\frac{\mathrm{Number}\;\mathrm{of}\;\mathrm{the}\;\mathrm{fungus}\text{-}\mathrm{infected}\;\mathrm{grains}}{\mathrm{The}\;\mathrm{total}\;\mathrm{number}\;\mathrm{of}\;\mathrm{tested}\;\mathrm{grains}\;}\times 100$$

Full morphological description of the isolated seed-borne fungi was done based on the cultural, morphological, and microscopic characteristics [27,28,29].

### 2.5. Pathogenicity Test

Eighteen fungal isolates affiliated with 9 species and 4 genera of maize were selected, as they are the most abundant in this study. The inoculum of each fungal isolate was started by growing them on PDA plates and incubating at 25 ± 2 °C for 5 days, and then, mycelium plugs of each fungus were used to inoculate a sterilized medium of sorghum:sand: water (2:1:2 *v/v*) and incubated at 25 ± 2 °C for 10 days. Plastic pots (20 cm in diam.) were stuffed with 4 kg per pot of disinfected soil. The soil was obtained from the Al-Qaseem Governorate, the soil was silt–sand–clay with a loamy texture, containing 3.38% coarse sand, 29.41% fine sand, 38.19 silt, and 29.02% clay, with 1.19 organic matters. The EC was 0.85 ds m^−1^ in the soil past, the pH in the water suspension = 7.9, and CaCO_3_ = 4.15. The available N, P, and K contents were 51, 4.55, and 138.4 mg kg^−1^, respectively.

The pots were individually inoculated with the earlier prepared fungal inoculum at the rate of 0.4% (*w/w*) and regularly irrigated with tap water close to field capacity and left for a week to verify the spread of the fungal growth. Control pots containing steam-sterilized soil watered by water only. All pots were organized in a randomized block design and saved in the greenhouse. Healthy seeds of maize seeds (cv. legami) were surface-sterilized using a NaClO solution (2%) for 3 min, washed with sterile water, and plot-dried on tissue paper. Ten grains per pot were planted with fifteen replicates per tested fungus. All pots were kept in a greenhouse environment (day temperature 25 ± 3 °C, night temperature 20 ± 3 °C, and 16-h photoperiod) up to 45 days. After 14 days, the percentages of the rotted grains, postemergence seedling damping-off (seedling’s infection percentages after 45 days from planting), and plants that survived were calculated.

### 2.6. Biodiversity Metrics

The biodiversity of the pathogenic fungal species recovered in the surveyed maize-cropping sites of KSA was calculated. The frequency (Equation (1)) and relative abundance (evenness, %) were determined using grain sampling and seed testing data aggregated across all 60 maize-growing sites. The relative abundance was estimated as: (3)$$\mathrm{Relative}\;\mathrm{abundance}\;\left(\%\right)=\frac{\mathrm{Number}\;\mathrm{of}\;\mathrm{grains}\;\mathrm{infected}\;\mathrm{with}\;a\;\mathrm{given}\;\mathrm{pathogenic}\;\mathrm{fungal}\;\mathrm{species}}{\mathrm{The}\;\mathrm{total}\;\mathrm{number}\;\mathrm{of}\;\mathrm{grains}\;\mathrm{infected}\;\mathrm{with}\;\mathrm{all}\;\mathrm{pathogenic}\;\mathrm{fungal}\;\mathrm{species}\;\mathrm{identified}}\times 100$$

The richness of the pathogenic species and the Shannon–Wiener diversity index (H) were determined for each of the 12 maize-growing governorates. The species richness was calculated as the total number of the pathogenic fungal species identified in a maize-growing governorate. The Shannon–Wiener diversity index was calculated using the following equation:(4)$$\mathrm{Shannon}–\mathrm{Wiener}\;\mathrm{diversity}\;\mathrm{index}\;\left(H\right)=-\sum_{i=1}^{s}\;p_{i}\times \mathrm{Ln}\left(p_{i}\right)$$
where pi = ni/N (ni is the number of grain samples with the species identified i, and N is the total number of samples with all fungus species identified), which is the relative abundance expressed in a fractional form, and Ln = is the natural logarithm. 

### 2.7. Biochemical Features of the Isolated Fungal Pathogens

#### 2.7.1. Cultivation Medium

The growth substrate of the solid-state fermentation medium was composed of one gram of a ground substrate (0.5 of maize grains and 0.5 g of straw) mixed with 5.0 mL of solution (1.6 g L^−1^ (NH_4_)_2_SO_4_, 4.0-g L^−1^ KH_2_PO_4_, and 1.0-g L^−1^ MgSO_4_) with pH 6 in 250-mL Erlenmeyer flasks, then autoclaved at 121 °C for 15 min. 

Each flask was inoculated with one 0.5-cm disk obtained from 7-day-old fungus grown on PDA. During the incubation period, the moisture content was kept at 65% by the addition of tap water when required. After incubation for 7 days at 28 °C, 10 mL of distilled water containing 0.01% Tween 80 was added to each flask, shaken on a rotary shaker (150 rpm for 30 min.), then filtered through Whatman No.1 filter paper [30], followed by centrifugation (5000 rpm for 15 min.). The filtrate was then examined for the various enzymes.

#### 2.7.2. Enzymatic Profile

Cellulases (filter paperase (FPase), carboxymethyl cellulase (CMCase), and β-glucosidase) activities were determined by incubating 0.5-mL enzyme and 0.5-mL buffer (0.05-M citrate buffer, pH 4.8) with 1% (*w/v*) cellulose microcrystal, carboxymethylcellulose, or cellobiose for 60, 30, and 15 min, respectively, at 50 °C. The reducing sugars released by FPase and CMCase were measured by the 3,5-dinitrosalicylic acid method (DNSA) [31]. The glucose units released by β-glucosidase were measured using the glucose oxidase kit (Spainreact Co., Girona, Spain). One unit (U) of CMCase, FPase, or β-glucosidase was defined as the amount of enzyme that releases one μmole of glucose per min under the assay conditions.

Polygalacturonases (PGase) activity was assayed by assessing the reducing groups discharged from 1% polygalacturonic acid in 0.1-M (pH 5.2) sodium acetate buffer at 40 °C for 30 min using DNSA. One PGase U was described as the enzyme amount that yields one μmole-reducing end (D-galacturonic acid monohydrate) g^−1^ min^−1^ under the assay situations [32].

The activity of α-amylase was evaluated in a mixture containing amylase and 1% soluble starch in a citrate phosphate buffer (pH 6.6) incubated at 40 °C for 30 min [33]. The released monosugars were measured by DNSA. One U of α-amylase was defined as the enzyme amount that released 1 μmole of glucose equivalent g^−1^ min^−1^ under the assay conditions.

The proteolytic activity was quantified in a crude extract using casein as a substrate [34]. One U of protease activity was expressed as the amount of the enzyme resulting in the release of 1 µg of tyrosine equivalent g^−1^ min^−1^ under the assay conditions.

For the determination of chitinolytic activity, chitinase was assayed using with 3-mg/mL chitin azure in 0.2-M potassium phosphate (pH 7) buffer at 30 °C; after 30 min of incubation, the mixture was boiled for 5 min, then centrifuged at 13,000× *g* for 5 min. The absorbance was assessed at 575 nm. One U of chitinase was the amount of enzyme that produced an increase of 0.01 in the absorbance under the assay conditions [35].

#### 2.7.3. Fungal Content of Protein and Amino Acids

The freeze-dried mycelium was used according to Christias et al. [36]. Extraction and estimation of the total protein from the fungal mycelia were done by grinding 40 mg in an electric mill using 2 mL of extraction buffer (0.2-M Tris–HCl (pH 6.8), 2% SDS, and 10% sucrose) and stored overnight at −20 °C, then centrifuged at 12,000 rpm for 10 min, and then, the supernatant was gathered. The total protein content of the mycelial extracts was assessed using the method of Bradford (Protein Biuret Method, www.bio-diagnostic.com, accessed on 15 May 2021). The protein content was analyzed using a UV spectrophotometer at 550 nm (U-3501 spectrophotometer (Hitachi Corporation, Tokyo, Japan). Bovine serum albumin was used as the standard protein. 

Extraction and quantification of the free amino acids of the fungal strain were done [36] with slight modifications. About 2 g of freeze-dried fungal mycelial powder was homogenized with 2 mL of boiling distilled water containing 5% trichloroacetic acid and 0.3-mL ethanol with continuous shaking for 15 min. The mixture was centrifuged at 10,000 rpm, and the supernatant was collected, then evaporated until dryness under reduced pressure at 45 °C.

Samples of the amino acid were spotted on the chromatography paper (Silica Gel 60 F254 TLC aluminum sheet 20 × 20 cm, thickness 0.1 mm; Merck, Germany) by graduated capillary tubes (5-μL volume; Spectrochem). TLC papers were air-dried and exposed to butanol:glacial acetic acid:water (12:5:3, *v/v*) as the mobile phase, then dried. The spot of every amino acid was accurately scraped for determination into test tubes within two hours after the chromatogram had been run [37].

### 2.8. Molecular Biology Relationships 

The 12 severe pathogenic fungi were selected for molecular identification. About 0.5 g. of the isolated fungal mat was subjected to DNA extraction using the DNeasy 96 Plant Mini Kit (QIAGEN, Hilden, Germany). DNA concentration and purity were measured by Nanodrop (SPECTROstarNano, Offenburg, Germany), and primers ITS1 and ITS4 were used for PCR amplification for fungal identification and seven RAPD primers for studying the genetic diversity among the pathogenic fungi (Appendix A). 

The PCR reaction mixture consisted of 1× buffer (Promega, Madison, WI, USA), 15-mM MgCl_2_, 0.2-mM dNTPs, 20 picomoles of each primer, 1 μL of Taq DNA polymerase (GoTaq, Promega), 40-ng DNA, and ultra-pure water to a final volume of 50 μL. PCR amplification was performed in a Perkin-Elmer/GeneAmp^®^ PCR System 9700 (PE Applied Biosystems, Waltham, MA, USA) programmed to fulfill 35 cycles after a preliminary denaturation cycle for 5 min at 95 °C. Each cycle consisted of a denaturation step at 95 °C for 30 S, an annealing step at 51 °C for 30 S, and an extension step at 72 °C for 30 S. The extension segment was extended for an additional 7 min at 72 °C in the final cycle. The amplification products were resolved by electrophoresis in a 1.5% agarose gel containing ethidium bromide (0.5 μg/mL) in 1X TBE buffer at 95 volts. Gels were visualized under UV light and photographed using a Gel Documentation System (BIO-RAD 2000, Dubai Branch, United Arab Emirates). The amplified DNA banding patterns were analyzed by Gel works Image Lab^TM^ Software (www.bio-rad.com, accessed on 15 May 2021). The polymorphism (%) for each primer was estimated using the following equation [38]:(5)$$\mathrm{Polymorphism}\;(\%)=\frac{\mathrm{Number}\;\mathrm{of}\;\mathrm{polymorphic}\;\mathrm{bands}}{\mathrm{Total}\;\mathrm{bands}\;}\;\times \;100\;$$

#### DNA Sequencing Analysis

The PCR products were purified using a QIAquick PCR purification kit (QIAGEN, Hilden, Germany), and the purified DNA was subjected to DNA sequencing. The sequencing of the product PCR was carried out in an automatic sequencer ABI PRISM 3730XL analyzer using Big Dye^TM^ Terminator Cycle Sequencing Kits in an ABI 3730xl sequencer (Microgen Company). The relationships among the sequences were analyzed using the ClustalW program (https://www.genome.jp/tools-bin/clustalw, accessed on 15 May 2021) (http://www.ncbi.nlm.nih.gov/BLAST, accessed on 15 May 2021). The DNA nucleotide sequences for the 12 fungal isolates were deposited in GenBank, and the accession numbers of the studied isolates were obtained. MEGA 10 software was used to construct the phylogenetic tree based on the UPGMA (unweighted pair group method with arithmetic mean) statistical method (https://www.megasoftware.net/, accessed on 15 May 2021).

### 2.9. Statistical Analyses

The obtained data were expressed as the mean ± standard deviation. The heatmap analysis was constructed using the TBtools package. Principal component analysis (PCA) and canonical correspondence analysis (CCA) were computed using PAST (ver. 4, Past Software, University of Oslo, Oslo, Norway). An UPGMA cluster dendrogram of the studied fungi was performed based on molecular data from a random amplified polymorphism DNA (RAPD) marker by SYSTAT (ver. 13.2, Systat Software, Inc., San Jose, CA, USA).

## 3. Results

### 3.1. Distribution of Maize Seed-Borne Mycobiota

Forty-one species belonging to 24 fungal genera were retrieved from the maize samples by using the SMB, DFB, and AP methods. The SMB, DFB, and AP differed in their fungal recovery efficiencies (Table 1). Generally, *Aspergillus flavus*, *Aspergillus niger*, *A. glucus*, *Fusarium verticillioides*, *F. proliferatum*, *Penicillium* spp., *Rhizopus stolonifera*, and *Trichoderma asperellum* were the most abundant, while *A. flavus*, *A. niger*, and *F. verticillioides* were dominant.

Using the SMB technique, 19 genera and 36 fungal species were obtained from maize seeds, while 11, 18 genera, and 21 and 26 species were recovered using the DF and AP techniques, respectively. The AP procedure effectively detected the seed-borne saprophytes, e.g., *A. flavus* (95.0%), *A. niger* (93.33%), *Nigrospora oryzae* (53.33%), and *R. stolonifer* (46.67%). On the contrary, the SMB technique enhanced the recovery of *Alternaria alternata* (20%), *A. glucus* (35.0%), *Cladosporium* spp. (18.33%), *Penicillium* spp. (41.67%), and *Trichoderma asperellum* (33.33%). *Fusarium verticilloides* and *F. proliferatum* were pathogenic fungi effectively detected in a higher range using the SMB technique (61.67 and 36.67%, respectively) compared to the DF and AP ones (25.0, 31.67 and 11.67, 15.0%, respectively). However, the *Curvularia lunata*, *F. incarnatum*, *F. chlamydosporum*, *Macrophomina phaseolina*, *Sarocladium zeae,* and *S. strictum* pathogens were detected at lower frequency percentages. Moreover, nine fungi: namely, *Aspergillus albicans*, *A. clavatus*, *A. nidulans*, *A. tamarii*, *Fusarium incarnatum*, *Geotrichium candidum*, and *Stemphylium* sp., were detected only using the SMB technique but not detected with the DFB or AP technique, while the DFB method was able to detect *Chetomium globosum* and *Melanospora* sp. However, the AP technique succeeded to recover *Cunninghamella* sp. and *Macrophomina phaseolina*. *Aspergillus flavus* and *A. niger* showed the highest occurrence, which may impact seed germination and greatly drop the seed quality.

### 3.2. Pathogenicity of Maize Grain-Borne Fungi

For the selection of the pathogenic fungi, a total of eighteen fungal isolates comprising four genera and nine species were selected, as they are the most widespread in the current survey and worldwide known as pathogenic fungi on maize. The data (Table 2) indicate that a significant dissimilarity was recorded for the disease parameters of maize seedlings due to soil infestation with different seed-borne fungal isolates. Among the twenty tested fungal isolates, *F. vertiillioides* KSU1M1-2 and KSU1M1-3, *F. proliferatum* isolates KSU1M2-2 and KSU1M2-3, *S. zeae* KSU2M2-1 and KSU2M2-2, *C. lunata* KSU3M1-1, and *A. alternata* KSU4M1-1 showed the highest percentages of rotted maize seeds (23.67 and 29.17, 22.0 and 20.83, 23.0 and 20.33, 18.33, and 20.0%, respectively).

On the other hand, the symptoms of seedling mortality caused by *S. zeae* (isolates KSU2M2-1 and KSU2M2-2), *F. proliferatum* KSU1M2-2, *F. vertiillioides* (KSU1M1-1 and KSU1M1-3), *C. lunata* KSU3M1-1, and *Alternaria alternata* KSU4M1-1 developed the highest infection percentages and, consequently, lowered the seedling survival percentages as compared with the control treatment (96%). Following up the growing-on-test four weeks later, the results indicated that most tested fungi that caused mild infection in maize seedlings were *F. verticillioides* (KSU1M1-4 and KSU1M1-5), *F. chlamydosporum* KSU1M3-1, *F. incarnatum* KSU1M4-1, *C. australiensis* KSU3M2-1, and *S. strictum* KSU2M1-2 in the form of a significant decrease in seedling survival as compared with the control (96.0%). The infected plants showed symptoms of wilting of the topmost leaves and brown vascular bundles in a lower portion of the stem. Disorganized vascular strands due to pathogen invasion were observed. This caused a very pronounced weakening of the stalk. The symptoms extended to affect the leaves, which showed necrosis, chlorosis, yellowing, and later, turned black. In some cases, a whitish fluffy colony was shown on seeds and around the base of the seedlings.

The infection of maize seeds by *Sarocladium* stalk rot, caused by *S. zeae*, was observed during this study. Infected seedlings showed symptoms of vascular browning and both foliar and stalk wilt. To the best of our knowledge, the isolation of the above-mentioned fungal pathogens associated with maize seeds is the first report in Saudi Arabia. Accordingly, the pathogenic fungi that caused ≥25% mortality in comparison to control were selected for the next trials.

### 3.3. Biodiversity of Maize Seed-Borne Pathogenic Fungi

The pathogenic seed-borne fungi in the maize samples varied among the studied governorates. The frequency and relative abundance of the pathogenic fungi isolated from the maize-growing governorates were calculated (Table 3). *Alternaria alternata*, *F. proliferatum*, and *F. verticilloides* were the most frequent species (100% for each), recording the highest relative abundances at 21.8, 13.9, and 25.5%, respectively. The lowest frequency was 33.3%, recorded by *Curvularia australiensis,* with a relative abundance of 4.2%. 

The biodiversity of the fungal pathogens in terms of richness and the Shannon–Wiener diversity index was estimated in each of the 12 governorates surveyed (Figure 2). Both parameters displayed a substantial dissimilarity in the fungal species studied. Najran was the richest in the number of the detected pathogenic species, whereas Al-Ahsaa was the poorest. The Shannon–Wiener diversity index recorded its maximum in Najran and Gazan, followed by Aseer, with species diversity of 0.280, 0.262, and 256, respectively. On the other side, the lowest Shannon–Wiener was observed in Al-Jouf and Tabuk at 0.170 each.

### 3.4. Lytic Activity of the Pathogenic Fungi

The lytic activity of the pathogenic fungi isolated from maize grains was screened in terms of their hydrolytic enzymes and is introduced in Table 4. There was a wide variation in the enzymatic activity according to the isolates and the enzymes. The cellulolytic activity of the pathogenic fungi was determined, measuring three main enzymes, i.e., FPase, CMCase, and β-glucosidase. Most fungal isolates did not show any CMCase activity, except *S. zeae* KSU2M2-2. However, the pathogens exhibited high activity of β-glucosidase and FPase at 1009.09 U (*C. lunata* KSU3M1-1) and 12.37 U (*Alternaria alternata* KSU4M1-1), respectively. Regarding PGase, the pathogenic fungi isolated from maize grains showed pectinolytic activity ranging from zero up to 9.31 ± 0.65 U, which was recorded by *S. zeae* KSU2M2-2. The isolated pathogenic fungi were found to have reasonable amylolytic, proteolytic, and chitinolytic activities. The highest activities of amylase, protease, and chitinase were 8.58 U (*F. verticillioides* KSU1M1-4), 131.16 U (*C. lunata* KSU3M1-1), and 22.80 U (*F. verticillioides* KSU1M1-4), respectively.

### 3.5. Amino Acids and Protein Profile

The cells of the twelve pathogenic fungi isolated from maize grains were investigated for the amino acids content (Table 5). Of the essential amino acids, *F. chlamydosporum* (KSU1M3-1) had the highest content of tryptophan, leucine, isoleucine, methionine, phenylalanine, and lysine essential amino acids at 1.6, 6.177, 6.84, 2.333, 4.73, and 5.217 µmol/100 mg dry wt. *C. australiensis* (KSU3M2-1) had the highest amount of histidine and valine at 0.97 and 4.12 µmol/100 mg dry wt., respectively, while the highest value of threonine was 3.887 µmol/100mg dry wt., represented in *F. verticillioides* (KSU1M1-3). Of the nonessential amino acids, *F. verticillioides* (KSU1M1-3) had the highest content of alanine, glutamic acid, and aspartic, with values of 5.423, 5.857, and 4.907 µmol/100 mg dry wt., respectively. *F. chlamydosporum* (KSU1M3-1) had the highest content of arginine and proline, with values of 3.64 and 6.433 µmol/100 mg dry wt., respectively. The highest amount of serine and glycine was 3.753 and 5.18 µmol/100 mg dry wt., recorded in *C. australiensis* (KSU3M2-1). The total amount of protein varied from 164.53 in *F. proliferatum* (KSU1M2-2) to 225.74 µmol/100 mg dry wt. in *F. verticillioides* (KSU1M1-3).

### 3.6. Correlation among Biochemical Features of the Pathogenic Fungi

The results from the hierarchical cluster analysis of the heatmap showed two types of cluster dendrograms: pathogenic fungal isolates dendrogram in a vertical position and biophysiological and metabolic variables in a horizontal position (Figure 3). Based on the fungi isolates dendrogram, there are two main distinct groups: the first outgroup includes *S. zeae* (KSU2M2-2), and the second group contains three subgroups. The first subgroup includes *A. alternata* (KSU4M1-1) and F. proliferatum (KSU1M2-2); this subgroup mostly produced fungi for the FPase enzyme. Subgroup two included *C. australiensis* (KSU3M2-1), *F. verticillioides* (KSU1M1-1), and *F. verticillioides* (KSU1M1-4), namely the lowest fungi-producing β-glucosidase enzyme. The third subgroup was clustered into two sub-subgroups. The first sub-subgroup included F. chlamydosporum (KSU1M3-1), namely the most fungi-producing essential amino acids. The second sub-subgroup included two clades: the first clade consisted of C. lunata (KSU3M1-1), namely the most fungi-producing proteinase enzyme, while the second clade included the most pathogenic fungi, evinced by the incorporation of F. verticillioides (KSU1M1-3), F. verticillioides (KSU1M1-2), and *S. zeae* (KSU2M2-1). The red color indicated high similarity between the studied fungi, while the blue color indicated a low similarity.

### 3.7. Principal Component Analysis

The PCA was performed on the biophysiological and metabolic variables to illustrate the grouping and association between the studied pathogenic fungi. As shown (Figure 4), PCA1 is responsible for 98.696% of the variance, and the PCA shares with only 0.891% variance. The results of the PCA showed the clear separation of fungi collected from the Al-Qaseem region (*C. lunata* KSU3M1-1 and *F. verticillioides* KSU1M1-2) and *F. chlamydosporum* KSU1M3-1 and *F. verticillioides* KSU1M1-3 collected from the Middle region in a separate group, and the rest of the fungi in the other group.

### 3.8. Canonical Correspondence Analysis

The relationship between the environmental weather variables and the incidence of maize pathogenic fungi was explored using CCA (Figure 5). The eigenvalues of the first and second axes of CCA were 0.012 and 0.000 with a variance of 98.88% and 1.12%, respectively. The eigenvalues are considered the best measure for the quality of the ordination and the strength of the species-environment relationship, while a small value of eigenvalues is the best indication for ordination stability. The diagram of CCA demonstrated the relative position of the twelve pathogenic fungi, along with the environmental variables (temperature, relative humidity, pressure, and wind speed). The results from the CCA indicated that the temperature weather variable was the most significant one, followed by the relative humidity, pressure, and wind speed. The length of the arrow refers to the most powerful variable weather, and the direction of the arrow points refers to the highest weather variable change. The ordination from CCA showed *F. verticillioides* KSU1M1-2, *F. verticillioides* KSU1M1-3, *F. proliferatum* KSU1M2-3, and *F. proliferatum* KSU1M2-2 located in the right quadrant of the plot and a positive correlation with the relative humidity, pressure, wind speed, and temperature variable, while the other seven maize pathogenic fungi located in the left quadrant of plot showed the lowest correlation with the weather variables in addition to KSU1M1-4 located at the top away from the other fungi and showed also a low correlation with the variables of the environment.

### 3.9. Molecular Studies

#### 3.9.1. Molecular Identification

The seed-borne pathogenic fungi of maize were identified molecularly using the ITS region. The accession numbers of the twelve pathogenic fungi that were retrieved from GenBank were MW553174, MW553173, MW422773, MW422770, MW405882, MW405883, MW405871, MW405884, MW405885, MW422779, MW422774, and MW422772 for *Alternaria alternata* KSU4M1-1, *C. australiensis* KSU3M2-1, *C. lunata* KSU3M1-1, *F. chlamydosporum* KSU1M3-1, *F. proliferatum* KSU1M2-2, *F. proliferatum* KSU1M2-3, *F. verticillioides* KSU1M1-1, *F. verticillioides* KSU1M1-2, *F. verticillioides* KSU1M1-3, *F. verticillioides* KSU1M1-4, *S. zeae* KSU2M2-1, and *S. zeae* KSU2M2-2. These data were introduced alongside the full morphological description in Appendix A.

The obtained DNA nucleotide sequences of the 12 fungal isolates were aligned; accordingly, the strains displayed 98–100% similarity with their nearby relative isolates in GenBank. The phylogeny cluster tree topologies of the identified strains are demonstrated in Figure 6. A genetic tree was constructed from the twelve fungal isolates obtained in the study, with five different fungal isolates that were selected from GenBank based on the similarities between them and the twelve isolates. The tree classified the 17 fungal strains into two main clusters. Cluster 1 contains the strain *Alternaria alternaria* (MW553174, outer group), and the remaining 16 strains were included in the second cluster. The second cluster was divided into two subclusters; subcluster one was divided into two main groups: group one included *S. zeae* (CPO- MH299323, Mexico) and *S. zeae* (MW422772), while the second group contained *S. zeae* (MW422774) and *S. zeae* (MT032469, China). The second subcluster was divided into two other groups: the first group included *C. lunata* (MW4224773 and MW533137), *C. lunata* (KT309032, Spain), and *C. buchlose* (KX139031, Iran). On the other hand, the second group included all the fusarium strains obtained in this study in relation to *F. verticlliodes* (MK264336, China).

#### 3.9.2. Genetic Diversity of Fungi Using RAPD

Data of the RAPD-PCR profiles for the studied fungal isolates are illustrated in Table 6 and Appendix A. The total bands from all seven primers were 130 bands with sizes varying from 135 to 2215 bp; the highest number of total bands, polymorphic bands, unique bands, effective multiplex ratio (EMR), and marker index (MI) were recorded by primer RAPD 3 at 26, 26, 12, 26, and 23.81, respectively. The lowest polymorphism information content (PIC), EMR, and MI were reported by primer RAPD 3 at 0.82, 10.00, and 8.25, respectively. The percentage of polymorphism (P%) was 100% in all primers except RAPD 6 at 94.12.

A Cluster dendrogram of the fungal isolates from maize seeds generated from the RAPD molecular markers is presented in Figure 7. The cluster classified the fungi into two main groups; the first group included the two *Curvularia* spp., and the second group was classified into two subgroups: the first subgroup included *A. alternata*, two *S. zeae* (KSU2M2-1 and KSU2M2-2), two *F. verticillioides* (KSU1M1-1 and KSU1M1-4), and *F. chlamydosporum* (KSU1M3-1). The second subgroup contained the other *Fusarium* isolates (*F. proliferatum* (KSU1M2-2 and KSU1M2-3) and *F. verticillioides* (KSU1M1-2 and KSU1M1-3)).

## 4. Discussion

The distribution of maize seed-borne mycobiota was explored by several seed health testing methods. Sixty maize seed samples collected from different locations in 12 provinces of Saudi Arabia were examined for the incidence and frequency of seed-borne fungi using the SMB, DFB, and AP techniques. Forty-one fungal species comprising saprophytic and pathogenic ones were detected, in which their percent infestation was variable according to the locations. Our results revealed that maize seeds were infected by varying degrees with several pathogenic fungi, like *F. verticillioides*, *F. proliferatum*, *Alternaria alternata*, *Drechslera maydis,* and *S. strictum,* which are the primary causal agents of pink ear rot, *Alternaria* leaf blight, Southern corn leaf blight, and wilt diseases in most world maize-growing areas [39,40,41,42].

Regarding the detection methods, SMB, AP, and DFB showed substantial variances in the frequencies of the recovered fungi. The SMB technique was superior in detecting the maximum number (36 species) of seed-borne fungi compared to the DFB (21 species) and AP (26 species) techniques. The obtained results are consistent with the findings of many studies of research that the SMB technique provided ideal conditions for the development of mycelial growth and spore formation of many Hyphomycetes [43].

*Fusarium verticillioides*, often in association with *F. subglutinans* and *F. proliferatum,* are the most destructive prevalent causal pathogens of *Fusarium* ear rot or pink ear rot disease transmitted by maize grains worldwide and are dominant in drier and warmer climates worldwide [44], causing a reduction in maize output by 10% typically and 30–50% in severely affected areas [12]. Mycotoxin’s accumulation in preharvest-infected plants or stored grains like deoxynivalenol, zearalenol, fusaric acid, and fumonisin yielded by such pathogens impair human health and animals, as well as support the fungal virulence in severely infecting seedlings of various maize genotypes [13,41]. Moreover, *A. niger* and *A. flavus* were the most dominant in all used seed health techniques, recording <80% frequencies for each. *Aspergillus* species, e.g., *A. flavus* and *A. niger* are well-known agricultural pathogens that cause high economic losses by reducing the quality of infected maize grain, especially when ear rot symptoms appear, which are commonly associated with the secretion of the fungal aflatoxins and ochratoxins, as well as less-prominent toxins like patulin [45]. 

Our findings revealed that maize seeds were infected by the *C. lunata*, *C. australiensis*, *F. incarnatum*, *F. chlamydosporum*, *S. zeae,* and *S. strictum* pathogens at lower percentages. The detection of the above-mentioned fungal pathogens of maize grains constitutes the first comprehensive report of seed-borne of maize in Saudi Arabia. The presence of so many fungal pathogens on corn grains in high proportions calls for the need for field studies for these and other pathogens. Public awareness of the need to study seed health and develop practical resistance methods appropriate to improve the quality of these grains should also be increased.

The isolated maize grain-borne fungi were subjected to a pathogenicity test. The growing-on test showed symptom similarities in the infestation treatments of *Fusarium* species, especially *F. verticillioides*, *F. proliferatum,* and *F. chlamydosporum,* which were rotted seeds and stunted and yellow seedlings. In this respect, *F. verticillioides* KSU1M1-3 and *F. proliferatum* KSU1M2-2 recorded the highest significant decreases in seedling survival (56.67 and 62.83%, respectively) as compared with the uninoculated control (96.0%). Similar findings were reported earlier on *F. verticillioides* and *F. proliferatum* as the most destructive abundant pathogens associated with maize grains worldwide [35,39]. Both pathogens can survive in infected maize seeds without causing apparent symptoms on seed tissues and subsequently transmitted systemically to growing seedlings through the stalk up to the ear, causing blights and rot of the root, stem, and ear in the next season [12]. 

Although *Acremonium zeae* Gams & Sumner (*Sarocladium zeae*) was previously described as the seed-borne causal agent of the black bundle and *Acremonium* stalk rot diseases on maize [46], the current study reported the susceptibility of maize infection by *S. zeae*, *C. lunata,* and *C. australiensis,* recorded for the first time in the studied area of Saudi Arabia. Our data declare the ability of *S. zeae* (KSU2M2-1 and KSU2M2-2), *C. lunata* KSU3M1-1, and *C*. *australiensis* KSU3M2-1 pathogens to develop symptoms on maize seedlings in the form of a significant decrease in seedling survival as compared to the uninfected control. 

*Sarocladium zeae* was previously described as producing extracellular hydrolases, e.g., cellulases, amylases, and proteases [47], which are reported to play a role in the host-pathogen interaction; these enzymes were also reported in our isolates. On the other hand, a recent study identified *S. zeae* as the beneficial endophyte in maize seeds [42]. This contradiction may be due to the presence of different strains having, or not, various pathogenic activities on plants. However, information about the role of endophytic *Sarocladium* species, including *S. zeae*, in maize is still unclear. 

*Curvularia lunata* was previously reported as a causal pathogen of *Curvularia* leaf spot disease of corn in the United States and China under hot–humid and tropical areas, causing significant yield loss [48]. *C. lunata* can produce a non-host-specific toxin, furanoid (methyl 5-(hydroxymethyl) furan-2-carboxylate), as the virulence factor that induces the development of symptoms on maize plants comparable to those of natural infection [49].

The results obtained revealed that *Alternaria alternata* KSU4M1-1 caused dramatic losses of seedling survival (67%) as compared with the control treatment (96%). Recently, *A. alternata* was reported to be among four causal agents of *Alternaria* leaf spot disease on maize seedlings [50]. Other studies have stated that the risk is not only due to the yield loss they cause but also due to reduction in the grain quality (discoloration) and nutritive value, in addition to an array of *Alternaria* mycotoxins that belong to three derivatives: tetramic acid, dibenzopyrone, and perylene, which play an important role in the pathogenesis and can inhibit the germination of maize as well [51,52].

The pathogenicity test concluded *C. australiensis* KSU3M2-1, *C. lunata* KSU3M1-1, and *S. zeae* (KSU2M2-1 and KSU2M2-2) as the first recorded in KSA. However, based on the pathogenicity test, the most severe pathogenic fungi were selected for the rest of the study.

The biodiversity of the fungal species in every maize crop governorate was investigated. The Shannon diversity index is usually applied as a popular metric used in ecology for estimating species diversity. The index considers the number of species living in a habitat (richness) and their relative abundance (evenness). Therefore, the distribution level of the fungi among the species and their abundance in the governorates could be observed. The current biodiversity metrics exhibited that the Najran governorate recorded the greatest species richness and diversity. The biodiversity and species abundance among the studied region may be due to the variability of the climatic conditions that were characterized by semi-arid to arid climates with moderate humidity and wind speed levels and relatively high temperatures, which aid in the growth and dispersal of a wide range of fungi. 

The 12 isolated pathogenic fungi exhibited lytic activity during growth on the fermented medium. The tissue structure of maize plants, which was utilized during the fermentation, is hard to degrade, composed mainly of lignin, cellulose, pectin, and starch, as well as protein in lower amounts. Such a complicated construction restricts the penetration of pathogenic fungi into plant tissues. Therefore, the cooperation of various lytic enzymes may be required for the pathogenesis process. That is why the various hydrolytic enzymes, i.e., cellulases (FPase, CMCase, and β-glucosidase); amylase; and proteinase, were assayed. The previous enzymes are reported to be involved in the pathogenicity process since these enzymes are known as cell wall-degrading enzymes (CWDE) that facilitate the penetration of the fungal hyphae to enter plant tissue [15,30]. The great disparity in the enzymatic profile among fungal isolates may be due to the type of strain, the ecological conditions, and the genetic variation.

Cellulose is the main natural polymer composed of glucose units. The overall lytic activity of cellulase enzymes leads to the catalyzation of cellulose to single glucose units. The intermediate step of hydrolysis is performed by endoglucanase that randomly cleaves the β-1,4-glucosidic bonds in the inner cellulose, which, in combination with the catalytic action of two other exoglycanase enzymes (cellodextrinase and cellobiohydrolase) and β-glucosidase, lead to the complete degradation of the cellulose skeleton into glucose units [30,53,54].

Virtually, pectin is present in all plants and contributes to the cell structure. It is composed of galacturonic acid monomers. Pectin-degrading enzymes have two major classes: esterases and depolymerases. PGase is the main heterogeneous pectinase catalyzing the hydrolysis of the pectin polymer into galacturonic acid by cleaving the α-1,4-glycosidic bond [32,55].

Starch is the main component of maize grains, which is decomposed by amylase, a starch-degrading enzyme. The lytic behavior of this enzyme is mediated by β-amylase and γ-amylase, leading to the hydrolysis of starch into glucose [56].

The nitrogenous part of the plant is mainly hydrolyzed by proteases that cleave the protein into peptides and amino acids [15]. Exopeptidases catalyze the terminal peptide bond, and endopeptidases cleave the nonterminal bonds between amino acids [57].

The combined action of various lytic enzymes leads to the maceration of plant tissue, which facilitates the infection route by the pathogen. Phytopathogenic fungi produce CWDE to penetrate the host tissues through the degradation of the components of the plant cell wall (physical barrier against pathogenic invasion) [15] and, therefore, may be essential for pathogenicity.

Chitinase activity was detected in the majority of the tested pathogens. Chitinases catalyze the hydrolysis of the 1–4 β-glycoside bond of N-acetyl-D-glucosamine in chitin that is present in the cell wall of fungi. Therefore, chitinase is used as a fungicide and insecticide agent [35,58]. The pathogenic fungus may use chitinase as an additional aid in the pathogenicity process by antagonizing the other beneficial fungi that are present on plants, which guarantee its sole presence on plant tissue and the success of the pathogenicity process. It is worth mentioning that the fermentation medium was free from the chitinase substrate, consequently indicating that chitinase is secreted constitutively by the tested fungi. Three mechanisms are supposed to manage the biosynthesis of the microbial enzymes, which may be induced, constitutive, or both. The induced (inducible) enzyme is one whose secretion is stimulated by a specific substrate (chitin), whereas the constitutive enzyme does not require the presence of the specific substrate [15]. This leads to the conclusion that the current pathogens may have more than one mechanism of chitinase secretion, representing an additional threat for plant infection. Collectively, the current pathogens have a complementary profile of hydrolytic enzymes, capable of causing a serious infection of maize plants. 

Regarding the amino acids and protein content, this is the first investigation that differentiated the essential and nonessential amino acids in the studied seed-borne fungi of maize, which is very important considering their pathogenicity degree and biological properties. As stated earlier, amino acids are an integral part of the hydrophilic–hydrophobic properties of fungal cell walls, and they are included in the structure of microbial organic substances [59]. 

In this study, the amount of essential and nonessential amino acids of the 12 fungi was diverse and related to the degree of infection capacity of the fungi and the resistance ability of the host. For example, *F. chlamydosporum* (KSU1M3-1) has the highest amount of most of the essential amino acids (tryptophan, leucine, isoleucine, methionine phenylalanine, and lysine) that were related to its lower pathogenicity, whereas the highest content of nonessential amino acids (alanine, glutamic acid, and aspartic acid) was observed in *F. verticillioides* (KSU1M1-3) with the highest percentage of pathogenicity and the highest content of threonine. Pathogenic fungi contain different amino acids supporting and facilitating the invasion of fungi and characteristics as an important impact on fungal growth and fungal pathogenicity, as well as induce and activate virulence traits such as hyphal morphogenesis, the formation of a biofilm, and formation of the capsule [60,61,62]. 

Exploring protein polymorphism has been a central approach in the studying of the classification, identification, and genetic variability of specific and subspecific taxa of fungi, and further, the pathogenic fungi are classified based on the total protein into (i) fungi with the highest protein, (ii) fungi with moderate protein, and (iii) fungi with the lowest protein [63,64]. In this study, *F. verticillioides* (KSU1M1-3), *F. verticillioides* (KSU1M1-2), and *S. zeae* (KSU2M2-1) have the highest pathogenicity related to the highest induction of total protein. To establish successful colonization, plant pathogens, including vascular wilt fungi, secrete effector proteins during the pathogen-host interaction [65,66], which are key elements for the pathogen virulence against plants and are particularly important during the biotrophic phase of infection [67]. The resistant plants are recognized by having a special resistance protein that stimulates the activation of induced defense response against effector proteins [66,67].

The current data on amino acids in relation to pathogenicity encourage performing additional studies to discover this point. However, the differences among the 12 tested fungi in this study are expected to be caused by the diversity of the species and strains, as well as the environmental conditions, the different geographical origins, and the growth conditions of the analyzed species.

In this study, *F. verticillioides* (KSU1M1-3), *F. verticillioides* (KSU1M1-2), and *S. zeae* (KSU2M2-1) had the highest pathogenicity related to the highest total protein. The analysis of the protein polymorphism has been an important approach in the classification and identification, as well as the genetic variability, of specific and subspecific taxa of fungi. Several authors have reported that the total proteins can be used as a criterion method for identifying species of the same genus, such as *Fusarium* species [63,64]. 

Plant pathogens, including vascular wilt fungi, secrete proteins during colonization to establish a successful pathogen-host interaction [65]. Microbial pathogens secrete effector proteins, which are recognized by a special group of resistance proteins that stimulates the activation of the induced defense response [66]. These effector proteins are key elements produced by fungal pathogens for their virulence against plants and are particularly important during the biotrophic phase of infection [67].

In this study, the total amount of amino acids in the mycelium of the individual fungal strains tested was diverse. The highest total content of bounded amino acids was related to the degree of infection and the resistance ability of the host. 

A pathogen requires amino acids (AA) to support its physiological functions, and alterations in the AA availability have remarkable effects on the growth of a pathogen and its expression of virulence factors [68]. Amino acid metabolism is crucial to pathogenicity. Metabolic adaptation to the microenvironment of the host is associated with fungal morphogenesis, cell wall remodeling, biofilm formation, stress responses, and commensalism, all of which influence the progression of infections. Metabolic adaptation is regulated by complex transcriptional networks, such as the general control of amino acid metabolism [69]. The utilization of arginine by both host and pathogen represents a metabolic bottleneck that is critical in determining the outcome of a pathogenic infection [70]. The highest contents of amino acids facilitate the invasion of fungi and characteristics as an important impact on the fungal pathogenicity and growth, as well as induce and activate virulence traits such as hyphal morphogenesis, formation of the biofilm, and formation of the capsule [60,61,62].

The heatmap was used to study the similarities and dissimilarities among fungal taxa. A hierarchical cluster heatmap is one visual method that can be used to clarify the associations between different parameters of samples, such as weather variables and the different pathogenic fungi, whereas the PCA is an unsupervised learning method that reduces the dimensionality of a dataset and extracts only the most important information for analysis [71]. Both heatmap and the multivariate PCA were performed for visualization of the relationships, similarities, and dissimilarities based on the distances between various parameters against the genotypes. In this respect, the result of the PCA separated fungi according to their biochemical and physiological parameters. Fungal isolates from the Al-Qaseem and Middle Regions were separated from the rest of the fungi. 

The ordination diagram of CCA shows that the environmental conditions are critical limiting factors for the biology of fungi, affecting their growth, reproduction, and pathogenicity in many ways. The cumulative percentage for the correlations between species and environmental factors for the first and second axes illuminated 98.88% and 1.12% of the total variance. These results indicated that the environmental factors had a great impact on the distribution patterns of the studied pathogenic fungi. The current data indicate that temperature was the most influential environmental variable, followed by humidity, pressure, and wind speed. 

Environmental conditions such as humidity, wind, pH, pressure, and temperature affect the survival, growth, pathogenicity, and infestation of mycoflora [72]. Saudi Arabia is located in a tropical area, which has a desert climate characterized by high temperatures with high humidity and low rainfall. Accordingly, the correlation between the environmental conditions and colonization of phytopathogenic fungi in maize grain in Saudi Arabia was studied in this investigation by using CCA. The CCA is a multivariate method used for the arrangement of different species along with environmental variables. Moreover, the CCA demonstrates the linear combination of weather variables with the distribution of fungal species, as well as investigates the response of different species to the environmental factors [73].

It was stated that climatic factors such as temperature, pH, and humidity not only influence the fungal distribution but also affect the growth, survival, and infestation of *Fusarium* species in maize [72]. Furthermore, many studies have indicated that *Fusarium* species differed in their optimal temperature requirement for growth that was observed between 25 and 29 °C [74,75]. 

On the other hand, the high temperature may kill the pathogens directly or weaken them by sublethal heat, which makes them unable to damage crops [76]. We can conclude that temperature can affect fungal growth by affecting the kinetics of the cellular enzymatic reactions and modulating the secretum of the cell [77]. We assume that the lethal temperature is what stops these reactions and may affect the growth of the fungus and its ability to cause infection. Not only that, but the temperature difference may lead to the emergence of mutagenic isolates that may resist the high temperatures and can cause disease after adapting to the environmental changes.

Additionally, the humidity level was found to be a critical limiting factor for fungus vitality, affecting their distribution, reproduction, and pathogenicity in many cases. In general, the high water content may be necessary for some types of fungi for spore germination, as it is required for the initiation of germination; for instance, temperature and relative humidity are important metrological variables affecting ear and stalk rot infection of maize by *Fusarium* spp. [78].

CCA showed a low correlation of the incidence of *A. alternata,* as well as the current newly detected pathogenic fungi in KSA (*C. australiensis, C. lunata,* and *S. zeae*) with air temperature, relative humidity, pressure, and wind speed compared to the other pathogens. Our results showed that dry conditions and moderate temperature were the critical factors enhancing the prevalence and distribution of the above-mentioned pathogens, which may be explained by their sensitivity to low oxygen conditions in wet soils. High humidity conditions can be appropriate for the presence of antagonistic saprophytic microorganisms, including fungi, yeast, and bacteria, that may reduce the pathogen’s survival potential [79], whereas the *Magnaporthiopsis maydis* DNA levels increased 10 times under drought-stressed corn buds compared to non-stressed plants [80].

*Curvularia lunata* is a foliar destructive causal pathogen of *Curvularia* leaf spot disease of maize related to subtropical and tropical, as well as sometimes temperate areas, of the world [81]. Previously, *C. lunata* and *C. inaequalis* were first reported as the causal agent of *Curvularia* leaf spot disease of maize in North China [82]. It has been reported that the frequency of seed-borne *A. alternata* and *S. strictum* pathogens in maize grains varies significantly in drought and heat climates in different growing areas of South Africa [14].

The optimum range of *C. lunta* was from 28 to 32 °C and the pH values from 4 to 12. Similarly, the optimum in vitro growth of the *C. lunata* pathogen is around 30 °C [83]. Generally, high temperature and humidity conditions were found to be favorites for the development of *Curvularia* leaf spot disease during the flowering to grain filling stages [84]. Under such conditions, the accumulation of *C. lunata* mycotoxin (e.g., furanoid-type toxin) may occur both in vitro and in the plant, which can lead to leaf lesions [49]. On the contrary, a strong relationship between the dry weather conditions in the field and the *C. lunata* success overwintering in infected leaves and becoming primary infection sources in the following season was reported [82].

Regarding the molecular biology relationships, the phylogenetic framework was explored to detect the fungal evolution and taxonomic studies. In this study, studying the biodiversity and molecular identification of 12 pathogenic fungi in maize in KSA using the ITS region revealed 12 isolates from different species of fungi like *Fusarium*, *Curvularia*, *Alternaria,* and *Sarocladium* species. The studied isolates were classified with their nearby relative isolates, which showed highly significant relations among the members of each species. The obtained DNA nucleotide sequences of the 12 fungi displayed 98–100% similarity with their nearby relative isolates in GenBank. The phylogenetic tree classified the fungal strains into clusters, one containing only *Alternaria alternaria* (MW553174, outer group). The second cluster included all the other fungal strains, including the Fusarium strains obtained in this study. Intraspecies variability was noted in the ITS regions, which may be due to variances between the phenotypic and molecular taxonomic structures, as reported for the *Fusarium* genera [85].

In the current study, the discriminatory power of the RAPD markers was determined using various parameters. RAPD3 demonstrated the highest efficiency, which was characterized by the highest values of MI (23.81), EMR (26.00), and the percentage of polymorphic loci (100%). The value of PIC ranges from 0 to 1, with values closer to 1 indicating higher polymorphisms, i.e., PIC > 0.5 is classified as highly informative, values of 0.25 < PIC < 0.5 are moderately informative, and PIC < 0. 25 are lowly informative [86]. The present PIC values ranged from 0.82 in RAPD 2 to 0.95 in RAPD 1, which was higher than the average PIC value (0.261) revealed from 17 RAPD primers used to study the genetic diversity of *F. fujikuroi* from maize [87]. Such results indicate that the reliability and significance of RAPD in the characterization of the populations’ diversity is high. Some authors even indicated a higher RAPD sensitivity in the diversity determination when compared to other molecular techniques, such as RFLP [88] or SSR [89]. What is more, RAPD has been widely used in the judgment of genetic similarity [90] or in the analysis of genetic diversity within a species [91] or between closely related species [92]. Therefore, RAPD markers are dominant and efficient in the detection of DNA-sequence polymorphisms of many loci throughout the genome. For example, within the *Fusarium* genus, RAPD has been applicable in the distinction between nonpathogenic and pathogenic isolates of *F. oxysporum* [93] or *F. verticillioides* strain differentiation, depending on the host plant [94].

It has been observed that the results obtained by RAPD-PCR when compared with those obtained from the ITS DNA sequences confirm each other. The similarity among the examined individuals within the same species reached more than 95%, which indicates the importance of these methods in the differentiation between individuals within the same species.

## 5. Conclusions

Summing up, the current study provides background information on the correlation among the biodiversity, pathogenicity, biochemical features, phylogenic associations, and scattering patterns of the pathogenic seed-borne fungi of maize within the growing provinces of Saudi Arabia. Additionally, it shows the relations between the incidence of seed-borne fungal pathogens and different weather variables. The isolation trial concluded that twelve out of the 41 fungal species are pathogenic. Four fungi, *C. australiensis* KSU3M2-1, *C. lunata* KSU3M1-1, *S. zeae* KSU2M2-1, and *S. zeae* KSU2M2-2, were the first recorded in KSA. The temperature and relative humidity were the most influential weather variables. The hierarchical cluster heatmap classified the pathogenic fungi based on the pathogenicity and biochemical features (lytic enzymes, protein, and amino acids) and reported *F. verticillioides* (KSU1M1-3), *F. verticillioides* (KSU1M1-2), and *S. zeae* (KSU2M2-1) in one cluster group as the most pathogenic fungi. The ordination of CCA separated the studied fungi, along with the climatic factors, into two groups. Both the RAPD-PCR and ITS DNA techniques confirm the high similarities among the examined individuals within the same species.

Studying the relationships between the occurrence of causal pathogen diseases and different weather variables can be useful in predicting the presence and spread of fungal causal diseases to uninfected areas. The presence and spread of such pathogens must be considered when evolving crop disease protection strategies, which should include studying the seed health and examination of the seed quality before cultivation. In addition, our results can be worthwhile for other maize-cropping countries that share the same climatic conditions as KSU.

## Figures and Tables

**Figure 1 plants-11-00829-f001:**
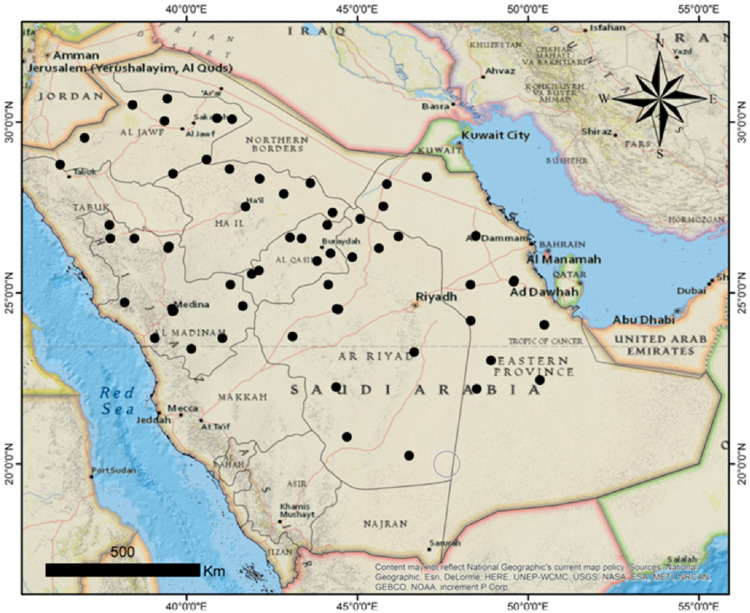
Map of sampling locations (black dots) showing the studied region in Saudi Arabia.

**Figure 2 plants-11-00829-f002:**
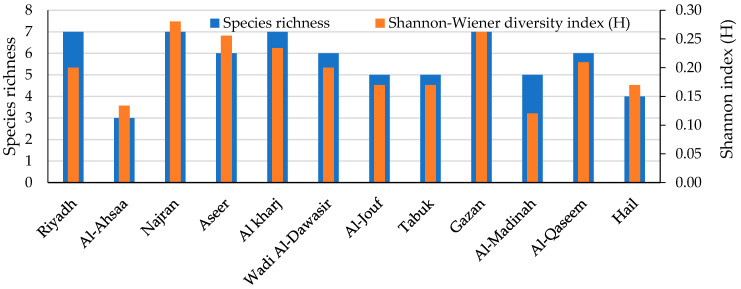
The richness and Shannon–Wiener diversity index of the 12 surveyed governorates.

**Figure 3 plants-11-00829-f003:**
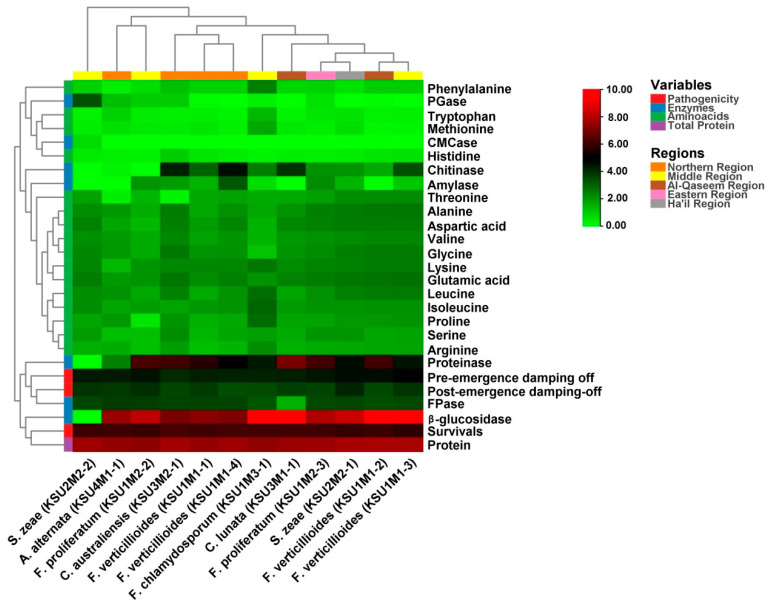
Heatmap showing the correlation between the biophysiological and metabolic parameters and the phylogenetic relationship among the twelve pathogenic fungi of maize in KSA. The pathogenic fungi collected from five regions are indicated by the upper bar in 5 colors, whereas the biophysiological parameters are indicated by 4 variables (enzymes, amino acids, pathogenicity, and total protein) by the left bar with 4 colors. The color scale denotes the variable level increasing from zero (deep green) to 10 (deep red).

**Figure 4 plants-11-00829-f004:**
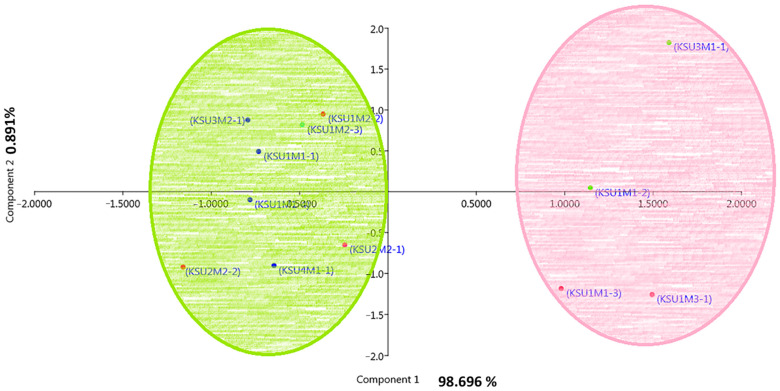
Principal component analysis of the biophysiological and metabolic variables of the 12 pathogenic fungi collected from KSA.

**Figure 5 plants-11-00829-f005:**
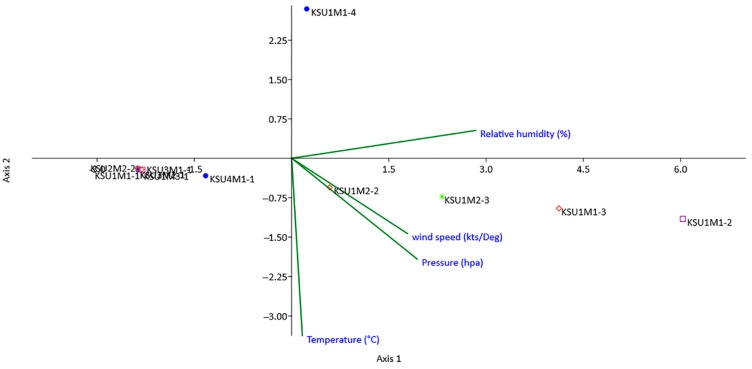
Diagram of the canonical correspondence analysis of the fungal isolates and environmental variables in KSA.

**Figure 6 plants-11-00829-f006:**
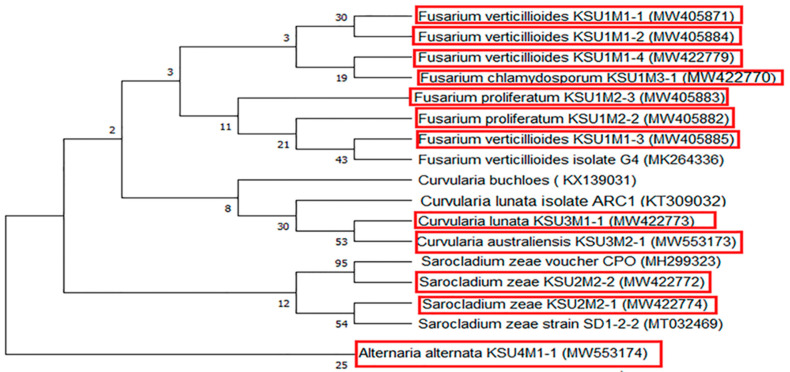
Phylogeny tree for the 12 isolated fungi from maize seeds in KSA inferred from ITS sequences (highlighted in red color) compared with other fungal isolates obtained from GenBank. Bootstrap tests were performed with 2000 replications.

**Figure 7 plants-11-00829-f007:**
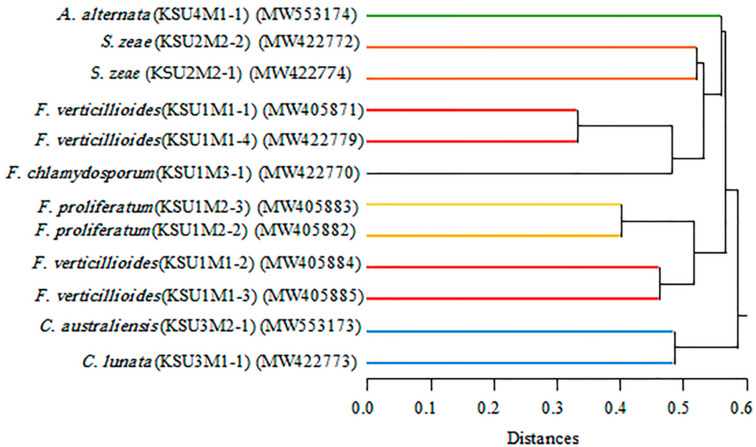
UPGMA cluster dendrogram of the twelve pathogenic fungi using the Euclidean distance average linkage method based on molecular data generated from seven primers of RAPD markers.

**Table 1 plants-11-00829-t001:** Occurrence of maize seed-borne fungi using the standard blotter (SMB), deep-freezing blotter (DFB), and agar plate (AP) methods.

Fungus	SMB		DFB		AP	
	F %	I %	F %	I %	F %	I %
*Absidia hesseltinii*	10.00	0.12 ± 0.062	0	0	3.33	0.024 ± 0.024
*Alternaria alternata*	20.0	0.42 ± 0.21	8.33	0.024 ± 0.024	15.00	0.220 ± 0.074
*Aspergillus albicans*	3.33	0.024 ± 0.024	0	0	0	0
*Aspergillus clavatus*	3.33	0.024 ± 0.024	0	0	0	0
*Aspergillus flavus*	85.0	39.02 ± 5.075	56.67	26.68 ± 4.13	95.0	32.61 ± 3.062
*Aspergillus fumigatus*	5.00	0.024 ± 0.024	6.67	0.049 ± 0.034	0	0
*Aspergillus glaucus*	35.0	1.61 ± 0.51	8.33	0.073 ± 0.041	0	0
*Aspergillus nidulans*	3.33	0.024 ± 0.024	0	0	0	0
*Aspergillus niger*	88.33	11.17 ± 1.30	53.33	3.73 ± 0.80	93.33	13.56 ± 1.65
*Aspergillus ochraceus*	16.67	0.17 ± 0.069	3.33	0.024 ± 0.024	0	0
*Aspergillus tamarii*	13.33	0.22 ± 0.11	0	0	0	0
*Aspergillus terreus*	20.00	0.46 ± 0.23	8.33	0.073 ± 0.041	10.0	0.12 ± 0.062
*Aspergillus* spp.	11.67	0.22 ± 0.13	5.00	0.098 ± 0.058	3.33	0.024 ± 0.024
*Cephalosporium acremonium*	13.33	0.41 ± 0.21	0	0	2.44	0.024 ± 0.024
*Cercospora* sp.	3.33	0.024 ± 0.024	0	0	0	0
*Chaetomium globosum*	0	0	8.33	0.073 ± 0.041	0	0
*Cladosporium* spp.	18.33	0.34 ± 0.049	10.0	0.17 ± 0.11	5.00	0.049 ± 0.034
*Cunninghamella* sp.	0	0	0	0	5.00	0.49 ± 0.034
*Curvularia australiensis*	5.00	0.46 ± 0.13	0	0	5.00	0.56 ± 0.20
*Curvularia lunata*	8.33	0.15 ± 0.089	0	0	6.67	0.024 ± 0.024
*Drechslera maydis*	3.33	0.024 ± 0.03	1.67	0.073 ± 0.04	3.33	0.049 ± 0.060
*Drechslera tetramera*	5.00	0.049 ± 0.034	3.33	0.024 ± 0.024	5.00	0.024 ± 0.024
*Exserohilum rostratum*	3.33	0.024 ± 0.024	0	0	3.33	0.56 ± 0.062
*Fusarium chlamydosporum*	15.00	0.15 ± 0.066	0	0	10.0	0.17 ± 0.085
*Fusarium incarnatum*	6.67	0.024 ± 0.024	0	0	0	0
*Fusarium proliferatum*	36.67	2.0 ± 0.61	11.67	0.049 ± 0.034	15.0	0.41 ± 0.20
*Fusarium verticilloides*	61.67	10.0 ± 2.84	25.0	0.27 ± 0.078	31.67	3.24 ± 1.49
*Geotrichium candidum*	3.33	0.049 ± 0.049	0	0	0	0
*Macrophomina phaseolina*	0	0	0	0	1.67	0.024 ± 0.024
*Melanospora* sp.	0	0	1.67	0.024 ± 0.024	0	0
*Neurospora* sp.	5.00	0.098 ± 0.077	0	0	8.33	0.56 ± 0.23
*Nigrospora oryzae*	23.33	1.44 ± 0.59	20.0	0.27 ± 0.092	53.33	3.68 ± 0.20
*Penicillium* spp.	41.67	1.66 ± 0.48	18.33	0.34 ± 0.15	25.0	0.66 ± 0.23
*Rhizopus stolonifer*	33.33	0.98 ± 0.26	13.33	0.24 ± 0.083	46.67	2.12 ± 0.49
*Rhizopus oligosporum*	10.00	0.85 ± 0.23	6.67	0.89 ± 0.20	21.67	3.2 ± 0.6
*Sarocladium strictum*	8.33	0.41 ± 0.13	5.00	0.024 ± 0.02	3.33	0.17 ± 0.20
*Sarocladium zeae*	5.00	1.60 ± 0.13	3.33	1.44 ± 0.20	0	0
*Stemphylium* sp.	6.67	0.073 ± 0.050	0	0	0	0
*Trichothecium roseum*	6.67	0.049 ± 0.034	0	0	5.00	0.024 ± 0.024
*Tricoderma asperellum*	33.33	0.32 ± 0.12	0	0	16.67	0.24 ± 0.024
*Ulocldium* sp.	5.00	0.049 ± 0.034	0	0	0	0

I % = incidence of a fungus; F % = frequency of the fungus.

**Table 2 plants-11-00829-t002:** Pathogenicity of maize seed-borne fungal species isolated *.

Fungus	Code	Preemergence Damping-Off (%)	Postemergence Damping-Off (%)	Survivals (%)
*Alternaria alternata*	KSU4M1-1	20.00 ± 1.00	13.00 ± 1.00	67.00 ± 1.73
*Curvularia australiensis*	KSU3M2-1	16.00 ± 1.15	11.00 ± 1.73	73.00 ± 2.52
*Curvularia lunata*	KSU3M1-1	18.33 ± 2.08	12.33 ± 1.53	69.34 ± 2.52
*Fusarium chlamydosporum*	KSU1M3-1	17.67 ± 2.52	10.33 ± 0.76	72.00 ± 2.18
	KSU1M3-2	9.330 ± 1.04	3.500 ± 0.50	87.17 ± 0.76
*Fusarium incarnatum*	KSU1M4-1	13.00 ± 1.00	9.17 ± 1.040	77.83 ± 0.76
*Fusarium proliferatum*	KSU1M2-1	13.83 ± 1.23	7.33 ± 0.58	78.84 ± 0.76
	KSU1M2-2	22.00 ± 2.00	15.17 ± 1.26	62.83 ± 2.93
	KSU1M2-3	20.83 ± 1.89	11.330 ± 1.32	68.17 ± 3.00
*Fusarium verticillioides*	KSU1M1-1	18.5 ± 0.50	12.83 ± 1.26	68.67 ± 1.15
	KSU1M1-2	23.67 ± 2.08	10.50 ± 0.50	65.83 ± 2.57
	KSU1M1-3	29.17 ± 1.04	14.17 ± 1.76	56.67 ± 1.90
	KSU1M1-4	17.67 ± 1.53	10.00 ± 1.00	72.33 ± 1.53
	KSU1M1-5	13.33 ± 1.53	8.830 ± 0.29	77.83 ± 1.61
*Sarocladium strictum*	KSU2M1-1	9.000 ± 1.00	7.000 ± 1.00	84.00 ± 1.73
	KSU2M1-2	13.67 ± 1.53	8.83 ± 0.29	77.50 ± 1.80
*Sarocladium zeae*	KSU2M2-1	23.00 ± 4.36	17.0 ± 3.00	60.00 ± 7.21
	KSU2M2-2	20.33 ± 3.21	14.0 ± 1.73	65.67 ± 2.52
Control (without infection)		3.330 ± 0.58	0.670 ± 0.58	96.00 ± 1.00

* Eighteen seed-borne fungal isolates were tested for their pathogenicity following a soil infestation method with fungus growing on a sterilized sorghum–sand medium (1:1) at 20% moisture for 10 d at 25 ± 2 °C. Pots (20-cm in diam.) stuffed with disinfected soil were individually infested with the fungal inocula at 0.4% (*w/w*), well-mixed, and regularly irrigated with tap water and left for one week before planting. In each pot, 10 surface-sterilized maize seeds were sown. For each fungus, 15 replicates (pots) were employed. All pots were organized in a completely randomized way and saved for 45 days in a greenhouse. The pots were noted daily for seed germination, and the disease occurrence was recorded. Each value is the mean of 15 pots (replicates), values within a column are followed by the mean ± standard deviation. Preemergence damping-off = seed/seedling death before emergence. Postemergence damping-off = seedling mortality after emergence. Survival = living plants 45 days after planting.

**Table 3 plants-11-00829-t003:** Frequency and relative abundance of the pathogenic fungi associated with maize grains in the studied governorates in Saudi Arabia.

Fungus	The Percentage of Appearance of the Fungus in Five Samples for Each Site												Total Frequency (%)	Relative Abundance (%)
	Riyadh	Al-Ahsaa	Najran	Aseer	Al kharj	Wadi Al-Dawasir	Al-Jouf	Tabuk	Gazan	Al-Madinah	Al-Qaseem	Hail		
*Alternaria alternata*	20	40	80	100	80	60	60	40	100	20	60	60	100	21.8
*Curvularia australiensis*	40	0	40	0	0	0	0	0	40	20	0	0	33.3	4.2
*Curvularia lunata*	40	0	60	60	40	60	0	20	80	0	40	0	66.7	12.1
*Fusarium chlamydosporum*	20	0	60	20	20	20	20	0	20	20	40	0	75.0	7.9
*Fusarium proliferatum*	40	40	60	60	20	20	20	40	60	20	40	40	100	13.9
*Fusarium verticilloides*	60	60	80	100	80	60	60	80	80	40	80	60	100	25.5
*Sarocladium zeae*	40	0	100	60	100	40	40	20	20	0	20	40	83.3	14.6

A total of 12 fungal pathogenic strains belonging to 7 species were chosen for the biodiversity metrics. They are *Alternaria alternata* (one strain), *Curvularia australiensis* (one strain), *Curvularia lunata* (one strain), *Fusarium chlamydosporum* (one strain), *Fusarium proliferatum* (two strains), *Fusarium verticilloides* (four strains), and *Sarocladium zeae* (two strains).

**Table 4 plants-11-00829-t004:** The activity of the hydrolytic enzymes of the pathogenic fungi isolated from maize grains.

Fungus	Enzyme (U)						
	FPase	CMCase	β-Glucosidase	PGase	Amylase	Proteinase	Chitinase
*Alternaria alternata* KSU4M1-1	12.37 ± 0.35	0.00	189.74 ± 3.42	1.44 ± 0.15	0.00	4.49 ± 0.09	0.17 ± 0.03
*Curvularia australiensis* KSU3M2-1	11.37 ± 0.43	0.00	132.43 ± 5.23	1.02 ± 0.03	2.82 ± 0.28	68.10 ± 0.76	18.28 ± 0.33
*Curvularia lunata* KSU3M1-1	1.87 ± 0.09	0.00	1009.09 ± 14.27	0.00	0.00	131.16 ± 6.51	14.50 ± 2.80
*Fusarium chlamydosporum* KSU1M3-1	8.74 ± 0.18	0.00	977.95 ± 27.15	0.20 ± 0.01	0.55 ± 0.05	20.32 ± 0.28	4.77 ± 0.15
*Fusarium proliferatum* KSU1M2-2	12.21 ± 1.05	0.00	289.65 ± 10.79	1.03 ± 0.04	3.29 ± 0.13	73.01 ± 2.05	0.00
*Fusarium proliferatum* KSU1M2-3	10.40 ± 0.16	0.00	246.04 ± 3.11	0.49 ± 0.07	3.79 ± 0.15	69.49 ± 1.57	3.40 ± 0.06
*Fusarium verticillioides* KSU1M1-1	11.14 ± 0.13	0.00	155.72 ± 3.11	0.05 ± 0.01	1.86 ± 0.05	52.78 ± 0.84	7.47 ± 0.26
*Fusarium verticillioides* KSU1M1-2	9.80 ± 0.79	0.00	847.14 ± 29.71	0.06 ± 0.01	0.00	67.45 ± 1.25	2.20 ± 0.06
*Fusarium verticillioides* KSU1M1-3	10.45 ± 0.49	0.00	787.96 ± 11.23	0.00	1.15 ± 0.13	21.30 ± 1.22	9.97 ± 0.03
*Fusarium verticillioides* KSU1M1-4	11.25 ± 0.14	0.00	138.36 ± 3.95	0.00	8.58 ± 0.28	32.18 ± 0.76	22.80 ± 0.58
*Sarocladium zeae* KSU2M2-1	11.00 ± 0.45	0.00	336.36 ± 19.45	0.00	1.77 ± 0.03	23.56 ± 0.80	3.20 ± 0.12
*Sarocladium zeae* KSU2M2-2	10.86 ± 0.38	0.64 ± 0.05	0.00	9.31 ± 0.65	0.00	0.00	0.00

**Table 5 plants-11-00829-t005:** Cellular amino acids and protein contents (µmol/100 mg dry wt.) of the pathogenic fungi isolated from maize grains.

Amino Acid	Aa (KSU4M1-1)	Ca (KSU3M2-1)	Cl (KSU3M1-1)	Fc (KSU1M3-1)	Fp (KSU1M2-2)	Fp (KSU1M2-3)	Fv (KSU1M1-1)	Fv (KSU1M1-2)	Fv (KSU1M1-3)	Fv (KSU1M1-4)	Sz (KSU2M2-1)	Sz (KSU2M2-2)
Essential amino acids (µmol/100 mg dry wt.)												
Histidine	0.217 ± 0.006	0.970 ± 0.020	0.270 ± 0.020	0.163 ± 0.015	0.193 ± 0.015	0.210 ± 0.030	0.410 ± 0.010	0.437 ± 0.015	0.497 ± 0.012	0.310 ± 0.010	0.310 ± 0.010	0.330 ± 0.010
Tryptophan	0.897 ± 0.015	0.217 ± 0.015	0.173 ± 0.015	1.600 ± 0.020	0.310 ± 0.030	0.420 ± 0.010	0.253 ± 0.012	0.190 ± 0.010	0.217 ± 0.006	0.167 ± 0.006	0.447 ± 0.006	0.180 ± 0.010
Leucine	3.400 ± 1.143	4.400 ± 0.010	2.430 ± 0.010	6.177 ± 0.006	2.340 ± 0.010	3.510 ± 0.010	2.223 ± 0.015	5.070 ± 0.010	5.113 ± 0.021	3.197 ± 0.015	4.580 ± 0.010	3.600 ± 0.020
Isoleucine	2.403 ± 0.025	2.450 ± 0.010	2.950 ± 0.010	6.840 ± 0.040	2.800 ± 0.010	3.350 ± 0.010	3.070 ± 0.010	3.457 ± 0.006	3.497 ± 0.012	3.257 ± 0.015	3.170 ± 0.010	3.710 ± 0.020
Methionine	0.523 ± 0.015	0.343 ± 0.006	0.350 ± 0.010	2.333 ± 0.021	0.353 ± 0.006	0.260 ± 0.010	0.417 ± 0.006	0.110 ± 0.010	0.127 ± 0.006	0.207 ± 0.006	0.570 ± 0.010	0.280 ± 0.010
Phenylalanine	0.250 ± 0.010	1.340 ± 0.010	0.760 ± 0.010	4.730 ± 0.010	0.600 ± 0.182	0.797 ± 0.015	0.830 ± 0.010	0.943 ± 0.006	0.983 ± 0.006	0.840 ± 0.010	0.473 ± 0.569	0.880 ± 0.010
Lysine	1.667 ± 0.015	4.080 ± 0.010	3.747 ± 0.006	5.217 ± 0.025	3.200 ± 0.010	4.250 ± 0.010	4.080 ± 0.010	4.890 ± 0.010	4.967 ± 0.059	4.110 ± 0.010	4.440 ± 0.017	4.230 ± 0.010
Threonine	0.243 ± 0.015	0.167 ± 0.015	2.287 ± 0.021	2.193 ± 0.015	1.707 ± 0.012	2.720 ± 0.173	2.340 ± 0.010	3.770 ± 0.010	3.887 ± 0.015	2.520 ± 0.010	3.823 ± 0.015	2.660 ± 0.010
Valine	2.900 ± 0.010	4.120 ± 0.010	3.070 ± 0.010	1.830 ± 0.010	2.180 ± 0.010	3.440 ± 0.010	2.660 ± 0.010	3.937 ± 0.015	3.983 ± 0.006	3.250 ± 0.010	3.760 ± 0.010	3.550 ± 0.010
Nonessential amino acids (µmol/100 mg dry wt.)												
Alanine	3.097 ± 0.015	4.880 ± 0.010	3.167 ± 0.015	2.317 ± 0.012	2.067 ± 0.031	4.673 ± 0.031	2.300 ± 0.010	5.307 ± 0.006	5.423 ± 0.015	3.510 ± 0.010	5.067 ± 0.059	3.630 ± 0.010
Serine	1.430 ± 0.010	3.753 ± 0.015	2.190 ± 0.010	2.690 ± 0.026	1.363 ± 0.015	3.160 ± 0.017	1.697 ± 0.015	2.433 ± 0.021	2.523 ± 0.025	2.427 ± 0.015	3.303 ± 0.021	2.870 ± 0.020
Arginine	2.140 ± 0.010	2.740 ± 0.010	1.953 ± 0.015	3.640 ± 0.020	1.380 ± 0.010	2.210 ± 0.010	1.440 ± 0.010	2.413 ± 0.015	2.460 ± 0.010	1.753 ± 0.015	2.313 ± 0.015	1.990 ± 0.020
Glutamic acid	2.980 ± 0.010	5.337 ± 0.006	4.240 ± 0.026	3.270 ± 0.010	3.203 ± 0.021	4.860 ± 0.020	3.700 ± 0.040	5.767 ± 0.006	5.857 ± 0.042	4.627 ± 0.021	5.247 ± 0.012	4.840 ± 0.020
Aspartic acid	2.413 ± 0.015	4.620 ± 0.010	3.927 ± 0.012	1.860 ± 0.020	1.650 ± 0.010	4.593 ± 0.021	2.840 ± 0.010	4.870 ± 0.010	4.907 ± 0.006	4.193 ± 0.021	4.850 ± 0.010	4.560 ± 0.020
Glycine	3.070 ± 0.010	5.180 ± 0.010	3.170 ± 0.010	1.220 ± 0.010	2.170 ± 0.026	3.793 ± 0.021	3.167 ± 0.015	5.040 ± 0.010	5.090 ± 0.010	3.417 ± 0.021	4.647 ± 0.025	3.980 ± 0.020
Proline	3.073 ± 0.006	3.347 ± 0.006	2.460 ± 0.010	6.433 ± 0.021	0.420 ± 0.026	2.523 ± 0.031	1.933 ± 0.031	3.133 ± 0.021	3.183 ± 0.006	2.300 ± 0.010	2.970 ± 0.020	2.450 ± 0.010
Total cellular protein (µmol/100 mg dry wt.)												
Protein	170.113 ± 0.021	211.077 ± 0.015	186.440 ± 0.020	166.450 ± 0.040	164.530 ± 0.020	193.867 ± 0.015	178.557 ± 0.031	225.180 ± 0.030	225.740 ± 0.061	199.195 ± 0.021	220.637 ± 0.025	199.830 ± 0.030

Aa = *Alternaria alternata*, Ca = *C. australiensis*, Cl = *C. lunata*, Fc = *Fusarium chlamydosporum*, Fp = *Fusarium proliferatum*, Fv = *Fusarium verticillioides*, and Sz = *Sarocladium zeae*.

**Table 6 plants-11-00829-t006:** Parameters generated from the RAPD markers.

Primer	Fragment Size (bp)	MB	UB	PB	TB	PIC	EMR	MI	RP	P (%)
RAPD 1	198–1150	0.00	9.00	25.00	25.00	0.95	25.00	23.74	2.53	100.00
RAPD 2	147–865	0.00	5.00	15.00	15.00	0.82	10.00	8.25	5.26	100.00
RAPD 3	140–1430	0.00	12.00	26.00	26.00	0.92	26.00	23.81	4.39	100.00
RAPD 4	180–1090	0.00	5.00	13.00	13.00	0.90	13.00	11.74	2.51	100.00
RAPD 5	265–2215	0.00	12.00	22.00	22.00	0.94	22.00	20.72	2.56	100.00
RAPD 6	135–1320	1.00	4.00	16.00	17.00	0.83	15.06	12.46	5.88	94.12
RAPD 7	135–1480	0.00	10.00	18.00	18.00	0.93	18.00	16.65	2.69	100.00
Total	135–2215	1.00	57.00	130.00	136.00	6.29	129.06	117.36	25.82	95.6%
Average		0.14	8.14	18.57	19.43	0.90	18.44	16.77	3.69	99.16

MB: monomorphic bands, UB: unique bands, PB: Polymorphic bands, TB: total bands, PIC: polymorphism information content, EMR: effective multiplex ratio, MI: marker index, RP: resolving power, and P (%): polymorphism percentage.

## Data Availability

The relevant data applicable to this research are within the paper.

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
