# Peer review of "A Case Study in Saudi Arabia: Biodiversity of Maize Seed-Borne Pathogenic Fungi in Relation to Biochemical, Physiological, and Molecular Characteristics"

_plants, 2022, doi:10.3390/plants11060829_

Round 1
Reviewer 1 Report
The manuscript entitled “Environmental conditions influence the biodiversity of maize seed-borne mycobiome in Saudi Arabia: the biochemical, physiological, and molecular attributes,” written by Abdulaziz et al., attempts to cover this gap to help forecast any destructive epidemics for the growing maize. From my point of view, this paper reports some interesting data about the maize seed-borne mycobiome and could be published after addressing some comments:
- In the abstract, define more clearly the problem you want to address via the considered environmental conditions, then bring the data and some in-depth outcomes.
- Please improve the introduction and bring more works about the environmental consequence of the biodiversity of maize seed-borne mycobiome. Additionally, get a table and compare your outcomes with previously reported data about environmental impact.
- Please add more in-depth data to the result and discussion section and clearly explain the environmental effect on target pathogens.
- The quality of images in Figure 5 is low. Please improve them.
- Authors should emphasize the practical novelty for such research and where are/would be applicable these obtained results, in Conclusion, for example.
Author Response
Thanks for your time and effort in the evaluation of the manuscript. We are enclosing herewith the revised manuscript. All comments raised from you were checked carefully. Hope the modifications meet all requirements.

Reviewer 2 Report
The revised manuscript is interesting and important with elements of novelty. Authors studied the mycobiome of corn in SA, which was new for this country.
By combination of molecular and biochemical studies they were able to identify the mycobiome and its physiological properties. This may be important to predict fungal effects of plant host.
For that reason I found the manuscript worth of publication in Plants MDPI Journal.
The aim is clear, the background is sufficient, methods are well describe. The collected results were statistically processed what makes readers believe the observations are genuine.
I have some comment for the improvement of the manuscript prior publication.
First of all I found many grammar and typing mistakes. The best would be to ask for help of any English native.
Here is a list mistakes I found:
L23: "Temperature and humidity were the most environmental variables influencing the fungal pathogenicity" - change into "Temperature and humidity were the environmental variables influencing the fungal pathogenicity the most".
L24: "The highest pathogenicity was also correlated with amino acids (threonine, alanine, glutamic acid, and aspartic), and protein" - what does it mean? Was the pathogenicity higher due to presence of given amino acids or their concentration? Please rephrase the sentence.
L34: better use America than Americans
L42: "...that are harmful to humans and animals, as well" change for e.g. "which are, in addition, harmful to humans and animals"
L52: "...was stated..." -> "was found to be linked to phatogenicity..."
L53: "vital task -> "vital role"
L55: "The fungal cell and its wall, in especial, support" -> "The fungal cell wall supports and protects..."
L56: "Protein in cell wall account (20-30%)" -> "Proteins constitute 20-30% of cell wall and are firmly..."
L61: "are representing" -> "represent"
L64-5: "..DNA sequencing for the identification of fungal groups and in comparison, of fungal biodiversity" -> "for description of fungal diversity"
L111, 144: sodium hypochlorite formula is NaClO
L123: "under stereoscopic and with help of a light microscope" - under light
supported stereoscopic microscope"
L135: common -> abundant
L136: "through its growing" -> "by sawing them ... and incubating..."
In general, writing units needs to be unified
In a case of temperature the symbol of degree is either underline or not - see lines 136, 138, 208, 236-238. Sometimes there is space between the number and degree and sometimes not.
The same applies for units, e.g. L195: 1umole -> 1 umole, L207: 2ml -> 2 mL L221: 5uL -> 3 uL. Decide to use one way of writing units.
L145: 'on A tissue paper"
L146: saved? -> kept in greenhouse?
L167: is -> was
L164 use subscript in (NH4)2...
L226: 5 gm. - use SI unit g
L231: 15mMMg... add missing space
L239: 0.5ug/ml missing space
Other parts and graphics used is well prepared. I cannot find supporting material.
KSA - it would be good explain once that it's Kingdom of Saudi Arabia
I wonder about origin of soils used in the point 2.5
What kind of liquid was used for irrigation?
There is missing name of sequencing project and the accession number of studied isolates. Were they introduced into database?
Author Response

(The authors gave the same response as above.)

Reviewer 3 Report
Lines 1-3 The title should change since it mentions environmental factors. Yes, they did not correlate any of the findings to environmental factors. So, the title is fully misleading. They only have used three different methods to isolate fungi from seeds of corn. Instead, all they have done is testing three methods for isolation of fungi.
Lines 20-21 The codes in the abstract are not necessary and should be removed.
Lines 98-99 The map of Saudi Arabia is uninformative. It does not show even the borders of the country or the states they talk about. It should change to a better map.
Line 205 – First they did not mention how they obtained mycelium and how dry or wet it was. Then I believe the determination of proteins and amino acid determination are totally irrelevant and should be removed and they did not provide any meaningful discussion as to why it was done.
Line 339 Table 3 is important but why they did not show all the isolates, and what are the numbers under each column like the column Riyadh. The title says Frequency and relative abundance of the pathogenic fungi associated with maize grains in the studied governorates. But under each state they have one number- so, which one is it Frequency or relative abundance? Completely unclear and not identified.
Table 1 Line 290 what is I% and F% why not defined?
This manuscript is not appropriate for publication at this form. But, I will be glad to review it again once extensive revision is made.
Author Response

(The authors gave the same response as above.)

Round 2
Reviewer 1 Report
The manuscript has been substantially improved and can be published in its current form. Congratulations on your effort!
Author Response
Deep thanks
Reviewer 3 Report
Please see attached doc for my comments

Author Response
Dear Reviewer 3
Thanks again for your time and effort in the evaluation of the manuscript. We are enclosing herewith the revised manuscript. All comments raised from you were checked carefully. Hope the modifications meet all requirements.
Thank you

Round 3
Reviewer 3 Report
Please see attached file

Author Response
Dear Reviewer
We are pleased with the valuable feedback on our manuscript. We admit that it added a good value to the research, which makes it good presented well than it was. Concerning the advice that there are some isolates with the same name, but they diverged from each other on constructed phylogenetic tree, this divergence was found when the isolates under study were compared with other isolates published on Genebank and they were closely related with ours but not identical. We have tried hard to get close to your point of view, so we have changed the phylogeny as much as we can, and now it was clear that most of the isolates under have same names became more closer to each other in the newly constructed phylogeny. Also, the figure legend was changed to clarify that we compared our 12 fungal isolates with the closely related isolates obtained from Genebank.
Thanks
